# A Feasibility Study of AlzLife 40 Hz Sensory Therapy in Patients with MCI and Early AD

**DOI:** 10.3390/healthcare11142040

**Published:** 2023-07-17

**Authors:** Sienna D. McNett, Andrey Vyshedskiy, Andrei Savchenko, Danijel Durakovic, George Heredia, Rael Cahn, Mikhail Kogan

**Affiliations:** 1Center for Integrative Medicine, George Washington University, Washington, DC 20052, USA; smcnett@gwu.edu; 2MET, Boston University, Boston, MA 02215, USA; vysha@bu.edu; 3Alzheimer’s Light, Miami, FL 33626, USA; support@alz.life (A.S.); pulzedev@gmail.com (D.D.); 4Department of Psychiatry and Behavioral Sciences, University of Southern California, Los Angeles, CA 90007, USA

**Keywords:** Alzheimer’s Disease, Mild Cognitive Impairment, light therapy, integrative medicine, technology, smart device, neurology, neuroscience

## Abstract

Alzheimer’s Disease (AD) and Mild Cognitive Impairment (MCI) are debilitating diseases that affect millions of individuals and have notoriously limited treatment options. One emerging therapy, non-invasive 40 Hz sensory therapy delivered through light and sound has previously shown promise in improving cognition in Alzheimer Disease (AD) rodent models. Small studies in humans have proven safe and tolerable, however exploration of feasibility and utility is limited. The purpose of this study is to examine the feasibility of this treatment in a human population through a smart tablet application that emits light and sound waves at 40 Hz to the user over the span of 1 h a day. Confirmation of entrainment of 40 Hz stimulation in the cerebral cortex was performed via EEG. 27 preliminary subjects with subjective cognitive complaints, Mild Cognitive Impairment, or AD were enrolled in the study; 11 participants completed 6 months of therapy. Of those that discontinued treatment, other health issues and difficulties with compliance were the most common causes. Participants were followed with Montreal Cognitive Assessment (MOCA) and Boston Cognitive Assessment (BOCA). For participants with subjective cognitive complaints, 2 of the 4 had improved MOCA score and 1 of 4 had improved BOCA score. For the participant with MCI, his MOCA score improved. For AD participants, 2 out of 6 had improved MOCA score and 3 of the 6 stayed stable, while 3 of 6 BOCA score improved. 4 of 11 participants specifically increased their MOCA scores in the Memory Index section. Of the 8 participants/caregivers able to speak to perceived usefulness of the study, 6 spoke to at least some level of benefit. Of these 6, 2 enrolled with subjective cognitive complaint, 1 had MCI, and 3 had AD. The therapy did not have reported side effects. However, those who did not finish the study experienced issues obtaining and operating a smart tablet independently as well as complying with the therapy. Overall, further exploration of this treatment modalities efficacy is warranted.

## 1. Introduction

Alzheimer’s Disease (AD) or Mild Cognitive Impairment (MCI) secondary to AD affect over 43.8 million individuals, with this number expected to continue to increase rapidly over the coming years [1]. Cognitive complaints associated with these conditions have a significant impact on the quality of life and psychological outcomes of patients and their caregivers [2]. Despite the devastating touch this disease has on individuals, families, the healthcare system and society, effective treatment options are severely limited [3]. This vacuity calls for further exploration in prevention, stabilization, and treatment of early AD and MCI.

On a cellular level, AD is characterized by insufficient amyloid beta (Aβ) plaque clearance with subsequent buildup of neurofibrillary tau tangles, neuronal dysfunction, and neural cell death [4]. While this pathological cascade is widely recognized, there are many gaps in understanding the link between these changes and clinical function. Due to this absence of thorough understanding, development of disease modifying treatments has been exceptionally challenging.

The call for novel emerging treatment options has prompted exploration of 40 Hz gamma wave therapy, which potentially holds disease modifying properties given its action on both functional and pathological characteristics of AD [5,6,7]. Brain activity operating at 30–100 Hz, are known as gamma oscillations [7]. These oscillations have been tied to both memory encoding and retrieval. Memory retrieval is disrupted in AD [7,8,9]. 40 Hz sensory stimuli has been shown to non-invasively induce these oscillations in rodents’ brains with disrupted memory retrieval [6,7].

Investigation of 40 Hz gamma wave sensory intervention in AD rodent models has shown promise. For one, 40 Hz light flicker treatment has been associated with decreasing amyloid load [6]. Furthermore, layering on a 40 Hz sound has been found to achieve increased cellular activation in brain areas related to memory and other cognitive performance including the cerebral cortex. Mice cognitive improvement in the domains of spatial and recognition memory with the auditory stimuli was found following 7 days of treatment [7].

Gamma wave entrainment therapy has been explored in human models to a limited extent at this time: a feasibility study by He et al. with 10 MCI patients used audiovisual 40 Hz stimulation for 4–8 weeks at 1hr a day and supported the tolerability, safety, and adherence of the therapy [10]. Another randomized control trial of 28 individuals with mild AD has shown improved cognition, biomarkers, and safety of the daily therapy after 3 months [11]. 

Of note, previous studies engaged new devices to emit light therapy, adding a layer of complexity and cost to treatment as they require obtainment and integration of an unfamiliar device into user’s routine. The introduction of a light therapy commercial tablet app improves accessibility and familiarity as it is delivered through a device that is already used in many households and has dual functions beyond just that of light therapy. Given the promise of this intervention and the paucity of disease modifying treatment options for MCI and early AD, further exploration of this non-invasive therapy with this alternate mode of delivery is warranted.

It is important to acknowledge and address the existence and limitations of existing pharmacological and non-pharmacological treatments of AD and why exploration of alternate/complementary therapeutic options such as the aforementioned sensory therapy is warranted. At this time, there are six FDA approved pharmacological treatments for AD; donepezil, rivastigmine, galantamine, memantine, Namzaric^®^ (memantine + donepezil), and finally the most recent, Aducanumab, which received accelerated approval by the FDA [12]. As in any pharmacologic intervention, these drugs each come with their own set of side effects ranging from mild to severe. The first three of the aforementioned pharmacological treatments are acetylcholinesterase inhibitors that target cognitive symptoms of AD, however lead to side effects such as nausea and diarrhea [13]. Memantine also has some neuroprotective properties, though also has side effects such as confusion and dizziness. These therapies have not been shown to extend disease modifying benefits [11]. Studies have shown Aducanumab as a disease modifying drug that works by decreasing Aβ plaque burden however is financially costly and has been tied to side effects such as dizziness, nausea, and in some cases Amyloid-related Imaging Abnormalities [14,15]. Studied non-pharmacological options have also thus far been demonstrated as largely symptomatic in benefit; ranging from physical exercise to memory training, however often require supervision/assistance by caregivers and can be limited by physical ability and other health concerns [15]. These side effects and limitations call for further exploration of alternative treatment options such as the aforementioned light therapy. It is the hope that a standalone or adjunctive therapeutic option may offer benefits or even disease modification properties beyond these existing therapeutics in a manner that is in some cases more financially and or physically more accessible to patients with a low side effect profile.

The goal of this feasibility study is to pilot 40 Hz light and sound sensory therapy using a novel smart tablet application in patients living with subjective cognitive complaints, MCI and neurologist diagnosed AD. Additionally, confirmation of stimulation of the cerebral cortex by the light and sound 40 Hz smart tablet application was evaluated via EEG to confirm the entrainment of the 40 Hz stimulation. The intervention employed by this study involved 60 min daily treatments implemented over a 6-month span with cognitive outcomes tracked and compared to established baselines. Compliance, safety, and subjective experiences of the therapy by both the participant and their caregivers were also explored when possible. Overall, the goal of the trial was to establish if further exploration of the efficacy of this intervention is warranted and feasible.

## 2. Materials and Methods

IRB approval was obtained through the George Washington School of Medicine and Health Sciences. (IRB#NCR191522). Participants were recruited through the GW Center for Integrative Medicine and GW University Memory Clinic, as well as through responses to an online bulletin posted on different Facebook groups and targeted emails. Exclusion criteria included history of epileptic or febrile seizure and macular degeneration or other significant disease that could impact delivery of the therapy through the eyes. The study was non-randomized and all enrolled participants were provided a free light therapy tablet application. 

### 2.1. Participants

All participants were required to provide their own iPad Pro device in a model capable of emitting 40 Hz light. The iPad Pro device was selected for this study as it was the first device capable of emitting the correct light frequency. At present, multiple other devices in iOS and Android systems are capable of emitting 40 Hz light frequency. Baseline cognitive testing consisted of Montreal Cognitive Assessment (MOCA) over telemedicine platform and Boston Cognitive Assessment (BOCA) on participant’s home device [16]. BOCA testing is a self-administered test that can be conducted through the iPad application with recent evidence as an effective cognitive assessment in a longitudinal clinical setting [17]. It was selected for this study as it presents the opportunity for regular cognitive follow up of light therapy at a regular interval without clinician weekly administration. 

Participants used the device for a total of 60 min each day for 6 months. Participants were permitted to break this 60 min into 20 or 30 min chunks throughout the day. Light flicker at 40 Hz was accompanied by 40 Hz sound generated by the AlzLife app and emitted by the iPad speakers, separate speaker, or over-the-ear headphones. Cognitive stimulation was provided by cognitive games presented over the flickering light screen using the same app. A choice of games included Sudoku, Tic-Tac-Toe, Clocks, Memory Games, Arithmetic, Checkers, Word Search, and over 30 other activities. Each activity had multiple levels of difficulty. Participants were instructed to play brain games of their choice at a level that is challenging but not overwhelming. Participants also had the option to use pure light therapy in their direct vision for the duration of treatment. Patients were instructed to stop therapy if they experienced any side effects and file a report with the research team. There was no compensation for participation except that all participants were provided a lifelong subscription to the AlzLife app. Instructions on use of the therapy were delivered over phone or video platform.

### 2.2. Evaluation of Induced Entrainment Using Scalp EEG Recording with Sensory Stimulation

To confirm entrainment of 40 Hz stimulation in the cerebral cortex, EEG was recorded in three cognitively normal young participants using a 64 channel NeuroscanSynamps II system, recorded at the sampling frequency 500 Hz. EEG was recorded for 15 min while participants engaged in the AlzLife app on a 2nd generation iPad Pro (10.5 inch, 2017 model) running AlzLife version 1.13.7(4), playing Sudoku with the “Maximum Light” and “Maximum Sound” settings selected. Additionally, a pre-stimulation and a post-stimulation baseline EEG was recorded for 3 min each with eyes open. Collected data were cleaned to remove artifactual segments due to scalp muscle contraction, head or neck movements, or excessive eye movements. The pruned data were divided into 2 s epochs and power spectral density (PSD) was calculated for each epoch. The total PSD was calculated by averaging all epochs for each condition.

## 3. Results

Entrainment of 40 Hz stimulation in the cerebral cortex was confirmed with scalp EEG in three participants. In all three participants AlzLife 40 Hz light and sound stimulation significantly increased the 40 Hz PSD relative to the baseline condition (Figure 1, solid black line for 40 Hz light and sound stimulation vs grey thin line for pre-stimulation baseline and dotted line for post-stimulation baseline). The 40 Hz peak was maximum in the frontocentral channels (FCz and Cz electrodes, Figure 1A). The 40 Hz peak was also detected in the occipital area (electrode POz, on the midline at the back of the head, Figure 1B).

A total of 57 individuals who were experiencing subjective or objective cognitive complaints were invited and 27 were enrolled in the trial (Figure 2). 11 participants completed the study, with the mean age of 80 years old with 9 females and 3 males. 4 of the participants had subjective cognitive complaints, 1 had neurologist diagnosed MCI, and 6 had neurologist diagnosed AD.

For participants with subjective cognitive complaints, 2 of the 4 had improved MOCA scores, Table 1. For the participant with MCI, their MOCA score improved as did 2 out of 6 AD participants with 3 of the 6 stayed stable. As such, only 1 of the 6 AD patients declined during the trial. 4 of 11 participants specifically increased their MOCA scores in the Memory Index section, Table 2.

As for BOCA scores, 4 participants increased in performance; 3 of these had AD and 1 had subjective cognitive complaints, Table 3. In addition to this quantitative testing we asked qualitative questions about perceived benefits. Of the 8 participants/caregivers able to speak to perceived benefits of the study, 6 spoke to at least some level of benefit, Table 4. Of these 6, 2 enrolled with subjective cognitive complaint, 1 had MCI, and 3 had AD. Below are several statements collected from both patients and caregivers:

Patients comments:P001 (enrolled with subjective cognitive complaints): *“Well I mean I can’t say it was like a lightning strike or something, but I sort of feel like it may have helped, maybe a little bit focusing, I don’t know how to describe it, just attention.”*P002 (enrolled with subjective cognitive complaints): *“I don’t have to search as long for words. [...] I find that I have regular conversations with people and I’m not caught in the middle of a sentence stumbling. I feel that that is better.”*P013 (AD): *“[The therapy] Improved my thinking. Could answer questions I couldn’t before in general conversation.”*P020: (AD) Has noticed *“Feeling more in control, not so frustrated”.*

Caregiver’s comments:P002: *Has noticed she “doesn’t get stuck in the middle of trying to find her words” like the “desk drawer is open”.*P003 (enrolled with subjective cognitive complaints): *“My impression is that there hasn’t been any deterioration.”*P023 (AD): *“During the first 3 months using the AlzLife app I thought she was showing improvement, I’m not sure now.”*

Of note, P024 had particularly advanced stages of AD and oftentimes was nonverbal per medical caregiver and family member. She was enrolled in the trial at the request and agreement of the family after making it clear to them that observable benefit will be very unlikely. The main reason to include this patient was to assess feasibility of administration of the treatment to patients with advanced AD.

Overall, participants in the light therapy trial found it tolerable and were able to integrate it into their daily routine without major difficulties or side effects. Compliance was reported as daily save the occasional missed day due to illness or travel for each of the individuals who completed the 6 months of therapy. Several participants noted that breaking the therapy session into shorter chunks throughout the day, and having the pure light therapy option in line of site while doing daily activities such as reading or listening to music made it fit easier into their daily routines. Albeit small, the improvement of MOCA and BOCA scores in some participants as well as reported subjective cognitive improvement in realms such as word finding are of note. Given the potential of cognitive benefits from this therapy method, the authors propose a randomized control trial of at least 50 or more individuals to further explore the utility of the therapy compared to placebo, which in this case would be use of the app without emission of 40 Hz frequency.

Benefits of this sensory therapy that emerged during the trial include that it can be administered at home without the aid of a medical professional, it is safe with very minimal health risk, and many participants said that it was fun and enjoyable. Obtaining and using the iPad Pro appears to be somewhat challenging for this group of patients given the initial financial cost and logistics of going to a store or ordering a device online. In future trials it would be warranted to provide participants with free devices to avoid this limitation. Furthermore, several participants were unfamiliar with how to use an iPad device and required detailed assistance in set up of the device by the research team or family members. This initial set up process was performed over telemedicine and the phone as the study was conducted over the COVID-19 pandemic, however could be streamlined in the future by having patients bring their device with them to be set up and receive an app tutorial in-office.

Given the large number of participants suspending therapy, we must consider possible contributing factors. First of all, many participants had difficulty with compliance did not have caregivers in the home able to provide daily prompting to complete the therapy. In contrast several participants that were able to complete the full 6 months had home health aids, spouses, or other family members who prompted their daily participation until a solid routine was established and accomplished by users. The sensory therapy fell by the wayside and either the participants did not respond to follow up or chose to suspend participation due to inability to commit to daily consistent use of the intervention. Several participants noted that a 60 min daily commitment was too much given other life and health commitments and chose to suspend. Also, participants that suspended therapy often had other health concerns and hospitalizations that overshadowed or interfered with therapy compliance. These factors must be considered when weighing the utility and feasibility of this therapy’s integration into a healthcare routine and introduce that lack of social support at home is a barrier to accessibility in some cases of this therapeutic intervention. Furthermore, it is possible that the loss of these participants to therapy completion may lead to results not representative of the general population, as the group who was able to complete the therapy may have had increased social support at home, less severe memory difficulties, and better overall health status.

Recruitment and follow up took place during the early COVID-19 pandemic, which complicated communication due to participant and potential participant contact and therapy set up being conducted over an online setting. Recruitment was limited to the online and telemedicine setting leaving out participants who did not have access or use these avenues of communication with care providers. It required participants or their caregivers to have access to the internet and an internet connecting device and operated video calling applications. Furthermore, many participants missed follow up appointments and cognitive testing appointments due to COVID-related illness, and some reported worsened mental health status during the course of the pandemic. If participants did not regularly check their emails, follow up cognitive testing and check ins with study administers were at times delayed and some patients were lost to follow up.

Missed data points, such as gaps in MOCA scores, were largely due to inconsistent patient follow up which was performed largely over email and telephone calls. It is notable that these inconsistencies were often secondary to unrelated patient illnesses and hospitalizations, travel, or lack of getting in touch with the participant’s family member. Delays in cognitive testing and unreported halts or interruption in therapy use may have limited possible cognitive improvements from the sensory therapy. Patients were encouraged to take cognitive testing only if they felt well and well rested, otherwise an alternate day was chosen.

While at this time evidence of treatment effectiveness is limited, feasibility is promising with the caveat that patient support in the home may be necessary to ensure proper compliance to the intervention. Alternately, a “reminder system” could potentially be developed to encourage compliance by users. This therapy would likely be best for patients with a system in place or caregivers who could help integrate the therapy into their routine. Continued check-ins with providers would be warranted to assess compliance. 

Moving forward from this feasibility trial, the next step in exploring this tablet based 40 Hz therapy would be to perform a double blinded study. Optimally, tablets would be provided to remove the potential financial hurdle to participation. Tablets would be unmarked, with half of participants receiving a device capable of emitting 40 Hz light and the other not capable. Both groups would use the AlzLife application for 60 min daily for 6 months. Cognitive testing would be performed regularly, with MOCA administered monthly, BOCA test self-administered weekly, and additional more in-depth cognitive assessments. Additional data involving presence of aid and social support in the home as well as if other cognitive, pharmacologic, or lifestyle interventions were being utilized, as well as presence of other active health issues and ADA status should be collected as well to explore for possible confounders to the data findings.

Overall, this innovative light and sound therapy treatment appears tolerable and safe with possibility of cognitive benefits. At the same time, barriers to utilization such as social support in the home and whether or not the participant already has access to/understanding of operating a smart tablet device vs would need to purchase and learn to operate one must be acknowledged. If these hurdles can be addressed by future studies providing access to these devices, allowing in office set up and teaching of how to operate the device, and the development of a way to prompt use of the therapy through a reminder system, these barriers are addressable. If proven beneficial in a randomized control trial it may become a key, non-therapeutic, and safe option in the prevention and or treatment of MCI and AD that is minimally limited by proximity to a healthcare location or physical ability versus existing interventions.

## Figures and Tables

**Figure 1 healthcare-11-02040-f001:**
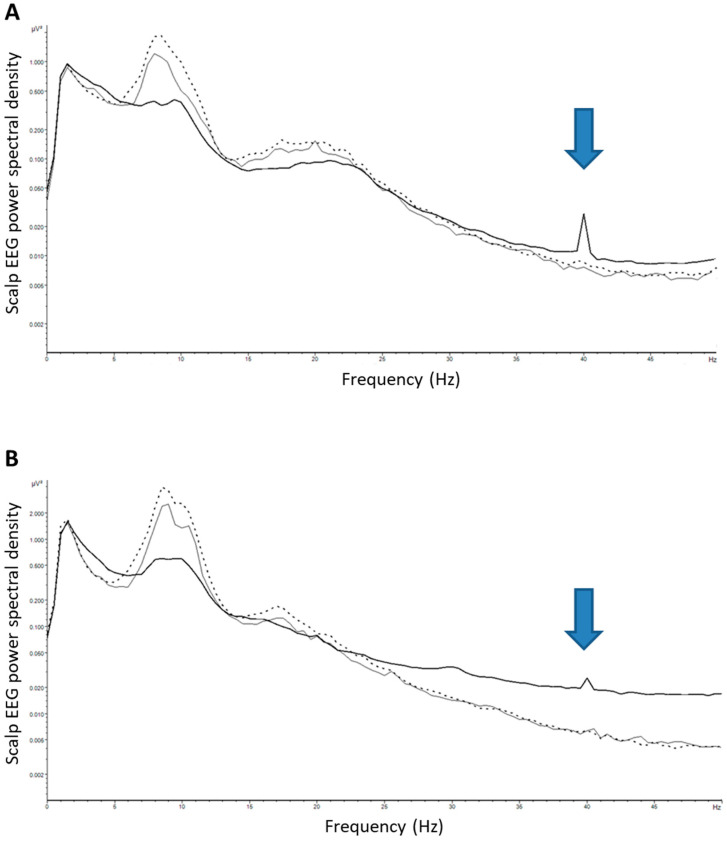
AlzLife 40 Hz light and sound stimulation entrains cortical regions. Scalp EEG power spectral density (PSD) at the frontal (FCz) electrode (**A**) and the occipital (POz) electrode (**B**) sites (*n* = 3). Solid black line: 40 Hz light and sound stimulation (active condition), group average; grey thin line: pre-stimulation baseline; dotted line: post-stimulation baseline. All 3 participants demonstrated the 40 Hz peak marked by an arrow.

**Figure 2 healthcare-11-02040-f002:**
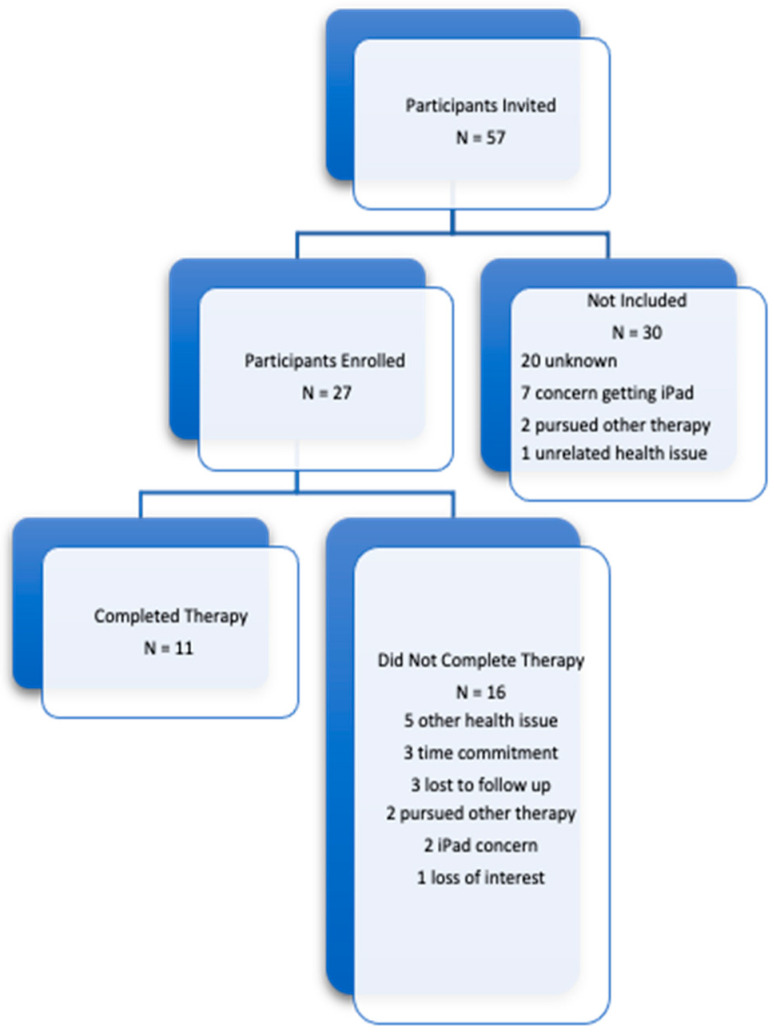
Enrollment Flowchart.

**Table 1 healthcare-11-02040-t001:** MOCA Scores.

	Sex	Age	Diagnosis	MOCA Base Line	MOCA 3 Months	MOCA 6 Months	MOCA 3 Months Change from Baseline	MOCA 6 Months Change from Baseline	MOCA Improvement
**P001**	F	72	Subjective cognitive complaints, family hx of AD	29	29	30	0	1	Y
**P002**	F	69	Subjective cognitive complaints, APOE4++ status	27	30	29	3	2	Y
**P003**	F	75	AD	23	21	24	−2	1	Y
**P004**	F	67	Subjective cognitive complaints, family hx of AD, APOE4+ status	30	29	28	−1	−2	N
**P005**	F	87	AD	22	-	23	-	1	Y
**P009**	F	75	Subjective cognitive complaints	26	25	25	−1	−1	N
**P0013**	M	95	AD	19	19	19	0	0	N
**P0014**	M	77	MCI	23	21	25	−2	2	Y
**P020**	F	82	AD	-	26	26	0	-	-
**P023**	F	89	AD	23	17	18	−6	−5	N
**P024**	F	89	AD	0	0	0	0	0	N
**Mean**							**−0.9**	**−0.1**	
**SD**							**2.2**	**2.0**	

**Table 2 healthcare-11-02040-t002:** MOCA Memory Index Scores. Higher scores indicate better cognition; the maximum score is 30.

	Sex	Age	Diagnosis	MOCA-MIS Baseline	MOCA-MIS 3 Months	MOCA-MIS 6 Months	MOCA-MIS 3 Months Change from Baseline	MOCA-MIS 6 Months Change from Baseline	MOCA-MIS Improvement from Baseline
**P001**	F	72	Subjective cognitive complaints, family hx of AD	14	14	15	0	1	Y
**P002**	F	69	Subjective cognitive complaints, APOE4++ status	11	14	13	3	2	Y
**P003**	F	75	AD	6	2	15	−4	9	Y
**P004**	F	67	Subjective cognitive complaints, family hx of AD, APOE4+ status	15	15	13	0	−2	N
**P005**	F	87	AD	8	-	8	-	0	N
**P009**	F	75	Subjective cognitive complaints	11	11	5	0	−6	N
**P0013**	M	95	AD	-	3	4	3	4	-
**P0014**	M	77	MCI	12	11	8	−1	−4	N
**P020**	F	82	AD	-	13	13	-	-	-
**P023**	F	89	AD	3	4	0	1	−3	N
**P024**	F	89	AD	0	0	0	0	0	N
**Mean**							**0.2**	**0.1**	
**SD**							**2.1**	**4.3**	

**Table 3 healthcare-11-02040-t003:** BOCA Scores. Higher scores indicate better cognition; the maximum score is 30.

	Sex	Age	Diagnosis	BOCA Baseline	BOCA 3 Months	BOCA 6 Months	BOCA 3 Months Change from Baseline	BOCA 6 Months Change from Baseline	BOCA Improvement
**P001**	F	72	Subjective cognitive complaints, family hx of AD	29	28	29	−1	0	N
**P002**	F	69	Subjective cognitive complaints, APOE4++ status	28	27	28	−1	0	N
**P003**	F	75	AD	16	18	21	2	5	Y
**P004**	F	67	Subjective cognitive complaints, family hx of AD, APOE4+ status	28	25	29	−3	1	Y
**P005**	F	87	AD	21	-	-	-	-	-
**P009**	F	75	Subjective cognitive complaints	25	-	-	-	-	-
**P0013**	M	95	AD	15	22	30	7	15	Y
**P0014**	M	77	MCI	-	-	-	-	-	-
**P020**	F	82	AD	26	24	28	−2	2	Y
**P023**	F	89	AD	21	18	-	-	-	-
**P024**	F	89	AD	0	0	0	0	0	N
**Mean**							**0.3**	**3.3**	
**SD**							**3.4**	**5.5**	

**Table 4 healthcare-11-02040-t004:** Subjective Improvement.

	Sex	Age	Diagnosis	Subjective Improvement (Participant)	Subjective Improvement (Caregiver)
**P001**	F	72	Subjective cognitive complaints, family hx of AD	Y	-
**P002**	F	69	Subjective cognitive complaints, APOE4++ status	Y	-
**P003**	F	75	AD	N	N
**P004**	F	67	Subjective cognitive complaints, family hx of AD, APOE4+ status	-	-
**P005**	F	87	AD	-	-
**P009**	F	75	Subjective cognitive complaints	-	-
**P0013**	M	95	AD	Y	Y
**P0014**	M	77	MCI	N	Y
**P020**	F	82	AD	Y	Y
**P023**	F	89	AD	-	Y then N
**P024**	F	89	AD	-	N

## Data Availability

De-identified raw data from this manuscript are available from the corresponding author upon reasonable request.

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
