# Peer review of "A Feasibility Study of AlzLife 40 Hz Sensory Therapy in Patients with MCI and Early AD"

_healthcare, 2023, doi:10.3390/healthcare11142040_

Round 1

Reviewer 1 Report

Dear author,

Thank you for the submission. This research described the feasibility and simple access of using the Alzlife 40Hz from App, however, the effect of this treatment is still unclear due to the very limited number of participants. There are some issues about this research that need to be revised and clarified. Please find the comments below.

1: Table 1. And Table 2 are cropped, please revise the size of the table.

2: The title of Table 4 is underlined which is not the same layout as other table titles, please revise to keep the consistency throughout these table titles.

3: In the 3. Results part line 103, as 27 out of 57 participants are patients and “of the 11 participants”, please provide more details about these 11 participants that completed the therapy, are they all MCI or AD patients? Or how many of them are normal elderly people?  Also, please provide MCI or AD participants’ medical records associated with AD, such as when were they diagnosed with MCI or AD.

4: Similar issue as “5 participants improved in MOCA score and 3 stayed stable”, “4 of 10 participants especially increase ”, please provide background info of these participants as well. Otherwise, it’s difficult to address the therapeutic effects on MCI participants.

5: Line 105, is it 4 of 11 participants or 4 of 10? 

Author Response

Edited paper has been attached thank you! I have increased the character count to above the minimum requirement and added more detail to the analysis and future directions in terms of how to analyze the efficacy of the therapy. 

1: Table 1. And Table 2 are cropped, please revise the size of the table.

Revised, hopefully they are the correct size and readable now 

2: The title of Table 4 is underlined which is not the same layout as other table titles, please revise to keep the consistency throughout these table titles.

Revised 

3: In the 3. Results part line 103, as 27 out of 57 participants are patients and “of the 11 participants”, please provide more details about these 11 participants that completed the therapy, are they all MCI or AD patients? Or how many of them are normal elderly people?  Also, please provide MCI or AD participants’ medical records associated with AD, such as when were they diagnosed with MCI or AD.

Revised, added in the MCI vs subjective cog complaint vs AD status. Unfortunately do not have access to exact length of each of their diagnostic status

4: Similar issue as “5 participants improved in MOCA score and 3 staye stable”, “4 of 10 participants especially increase ”, please provide background info of these participants as well. Otherwise, it’s difficult to address the therapeutic effects on MCI participants.

Revised, added this aspect into the results section 

5: Line 105, is it 4 of 11 participants or 4 of 10? 

Revised, 4 of 11 

Reviewer 2 Report

The manuscript provides a comprehensive overview of Alzheimer's Disease (AD) and Mild Cognitive Impairment (MCI) and their significant impact on the quality of life and psychological outcomes of patients. The author highlights the lack of effective treatment options and calls for further exploration in prevention and treatment of early AD and MCI. The introduction also provides a brief background on the cellular mechanisms underlying AD and the current pharmacological treatments available, including Aducanumab. The author then discusses the potential benefits of non-pharmacological options, such as gamma wave therapy, which has been shown to hold disease-modifying properties. The author also discusses the use of 40Hz sensory stimuli to induce gamma oscillations in rodents' brains with disrupted memory retrieval and the promising results of gamma wave entrainment therapy in human models. The author proposes a feasibility study to pilot 40Hz light and sound sensory therapy using a novel smart tablet application in patients living with MCI and early AD.

In summary, the article provides a promising glimpse into the potential benefits of light therapy for AD patients, but there are several areas where more detail and scientific rigor would improve the quality of the study.

Recommendations:

- Abstract is too short. Please rewrite abstract section.

- The introduction provides a thorough overview of the topic, but it could benefit from some reorganization to improve the flow and coherence of the ideas presented. In particular, the author may consider presenting the potential benefits of non-pharmacological options, such as gamma wave therapy and the use of 40Hz sensory stimuli, earlier in the introduction to provide context for the proposed feasibility study. Additionally, the author may consider providing more detail on the limitations and challenges of current pharmacological treatments and non-pharmacological options to highlight the need for alternative approaches.

- The article states that "further standardized and random controlled trial exploration are warranted to explore the benefits of this therapy." While this is certainly true, it would be helpful to specify what kind of study design and sample size would be necessary to establish the efficacy of the therapy. Additionally, the article mentions that missed data points were largely due to inconsistent patient follow-up, but it does not elaborate on how this might affect the validity of the results.

- The manuscript suggests that the therapy is easily accessible and able to be administered at home, but it also notes that obtaining and using an iPad Pro was challenging for some patients. It would be helpful to provide more details on what specific challenges patients faced and how they were addressed, as well as whether there were any other potential barriers to accessing the therapy.

- Lastly, the article acknowledges that the study took place during the early COVID-19 pandemic, which may have impacted the results. However, it does not provide any specific details on how the pandemic affected patient recruitment, therapy administration, or follow-up. Including this information would help readers better understand the context of the study and the potential implications of the findings.

Author Response

he manuscript provides a comprehensive overview of Alzheimer's Disease (AD) and Mild Cognitive Impairment (MCI) and their significant impact on the quality of life and psychological outcomes of patients. The author highlights the lack of effective treatment options and calls for further exploration in prevention and treatment of early AD and MCI. The introduction also provides a brief background on the cellular mechanisms underlying AD and the current pharmacological treatments available, including Aducanumab. The author then discusses the potential benefits of non-pharmacological options, such as gamma wave therapy, which has been shown to hold disease-modifying properties. The author also discusses the use of 40Hz sensory stimuli to induce gamma oscillations in rodents' brains with disrupted memory retrieval and the promising results of gamma wave entrainment therapy in human models. The author proposes a feasibility study to pilot 40Hz light and sound sensory therapy using a novel smart tablet application in patients living with MCI and early AD.

In summary, the article provides a promising glimpse into the potential benefits of light therapy for AD patients, but there are several areas where more detail and scientific rigor would improve the quality of the study.

Thank you very much for your input and recommendations! We have worked on adding additional data and detail and bolstered the length of the publication to above the minimum requirements. 

Recommendations:

  • Abstract is too short. Please rewrite abstract section. Rewritten and lengthened. 
  • The introduction provides a thorough overview of the topic, but it could benefit from some reorganization to improve the flow and coherence of the ideas presented. In particular, the author may consider presenting the potential benefits of non-pharmacological options, such as gamma wave therapy and the use of 40Hz sensory stimuli, earlier in the introduction to provide context for the proposed feasibility study. Additionally, the author may consider providing more detail on the limitations and challenges of current pharmacological treatments and non-pharmacological options to highlight the need for alternative approaches.

Have addressed by reorganizing the flow of the explored interventions, briefly added side effects and limitations of existing pharmacological and lifestyle interventions.

  • The article states that "further standardized and random controlled trial exploration are warranted to explore the benefits of this therapy." While this is certainly true, it would be helpful to specify what kind of study design and sample size would be necessary to establish the efficacy of the therapy. Additionally, the article mentions that missed data points were largely due to inconsistent patient follow-up, but it does not elaborate on how this might affect the validity of the results.

Have added additional detail in proposed future study design to examine utility of the therapy and expanded on the possible effect of the missed data points. Please let me know if this is properly addressing this comment appropriately. 

  • The manuscript suggests that the therapy is easily accessible and able to be administered at home, but it also notes that obtaining and using an iPad Pro was challenging for some patients. It would be helpful to provide more details on what specific challenges patients faced and how they were addressed, as well as whether there were any other potential barriers to accessing the therapy.

Have clarified challenges associated with access and use of the technology and barriers in place. 

  • Lastly, the article acknowledges that the study took place during the early COVID-19 pandemic, which may have impacted the results. However, it does not provide any specific details on how the pandemic affected patient recruitment, therapy administration, or follow-up. Including this information would help readers better understand the context of the study and the potential implications of the findings.

Have went into clearer detail on the challenges posed by the covid19 pandemic. 

Round 2

Reviewer 1 Report

Dear author,

Thanks for the revision, I have no further questions.